# Spatial patterns and predictors of missing key contents of care during prenatal visits in Ethiopia: Spatial and multilevel analyses

Aklilu Habte Hailegebireal[1]*, Habtamu Mellie Bizuayehu[2], Yordanos Sisay Asgedom[3], Jira Wakoya Feyisa[4]

1 School of Public Health, College of Medicine and Health Sciences, Wachemo University, Hosanna, Ethiopia, 2 School of Public Health, Faculty of Medicine, The University of Queensland, Brisbane, Australia, 3 Department of Epidemiology, College of Health Science and Medicine, Wolaita Sodo University, Wolaita Sodo, Ethiopia, 4 Department of Public Health, Institute of Health Sciences, Wollega University, Nekemte, Ethiopia

* akliluhabte57@gmail.com

**Data Availability Statement:** The data for this study was obtained from the DHS office upon a reasonable request, with the authorization to use it.

## Abstract

### Background

Quality Antenatal Care (ANC) is considered if pregnant women have access to essential services that align with the best evidence-based practice. Although several studies have been conducted on ANC uptake in Ethiopia, they have focused on the timing and number of visits and the level of complete uptake of care contents according to the WHO recommendation remains scarce. Hence, this study aimed to assess the magnitude of missing care content during ANC visits, its spatial variations, and individual- and community-level determinants in Ethiopia.

### Methods

The study was conducted using the 2016 Ethiopian Demographic and Health Survey and included a total weighted sample of 4,771 women who gave birth within five years before the survey. Spatial analysis was carried out using Arc-GIS version 10.7 and SaTScan version 9.6 statistical software. Spatial autocorrelation (Moran's I) was checked to determine the non-randomness of the spatial variation in the missing contents of care. Multilevel multivariable logistic regression analysis was performed using STATA version 16. The adjusted odds ratio (aOR) with its corresponding 95% CI was used as a measure of association.

### Results

The prevalence of missing full contents of ANC in Ethiopia was 88.2% (95% CI: 87.2, 89.0), with significant spatial variations observed across regions. Missing essential contents of care was higher among women who live in rural areas (aOR = 1.68, 95% CI: 1.47, 2.71), not completed formal education (aOR = 1.94, 95% CI:1.24, 3.02), late initiation of ANC (aOR = 3.05, 95% CI:1.59, 6.54), attended only one ANC (aOR = 4.13, 95% CI: 1.95, 8.74), and not having a mobile phone (aOR = 1.44, 95% CI: 1.07, 1.95).

The authors were granted the right to use the data solely for study purposes, with the condition that the data not be shared with any other third parties. Thus, the one who needs the data supporting the findings of this study can get it in anonymised form from the DHS website at https://dhsprogram.com/data/dataset/Ethiopia_Standard-DHS_2016.cfm?flag=1 upon reasonable request.

**Funding:** The author(s) received no specific funding for this work.

**Competing interests:** The authors have declared that no competing interests exist.

**Abbreviations:** ANC, Antenatal Care; aOR, Adjusted Odds Ratio; CSA, Central Statistical Agency; EDHS, Ethiopian Demographic and Health Survey; FANC, Focused Antenatal Care; WHO, World Health Organization.

## Conclusion

The level of missing care content during prenatal visits was high in Ethiopia, with significant spatial variation across regions. Health systems and policymakers should promote early initiation and encourage multiple visits to provide optimal care to pregnant women. In addition, it is vital to focus on enhancing education and healthcare infrastructure in rural parts of the country.

## Introduction

In 2020, an estimated 287,000 women died from maternal causes globally, equivalent to 800 maternal deaths per day, or one every two minutes [1]. Sub-Saharan Africa has the highest Maternal Mortality Ratio(MMR) (545 per 100,000 live births), which is more than two times the global estimate (223 per 100,000 live births) [1]. The region alone accounted for almost 70% of the global maternal mortality in 2020 [1, 2]. According to a World Bank modelled prediction in 2020, Ethiopia's MMR was 267 per 100,000 live births, which was higher than the global average of 223 deaths per 100,000 live births [3].

The primary causes of maternal death in Ethiopia, are obstructed labor, uterine rupture, hypertensive disorders of pregnancy, infection, and hemorrhage [4, 5]. Such causes can be prevented by adequate utilization of prenatal, intrapartum, and postpartum care [6]. For instance, high-quality antenatal care (ANC), defined as getting an adequate amount of visits and items of care on time, is one of the cost-effective approaches to improving MMR [7, 8].

ANC primarily aids women and newborns in maintaining normal pregnancies by identifying pre-existing problems and preventing complications during pregnancy and childbirth [9]. The extent to which pregnant women have access to essential services that fit with evidence-based professional knowledge as part of Antenatal Care (ANC) is an indicator of the quality of care during pregnancy [9]. To be effective, important ANC components must be offered to the ANC plan [10].

The contents of care are focused on the prevention of pregnancy complications (counseling on danger signs), prevention of iron deficiency anemia(iron supplementation), prevention of neonatal tetanus (via at least two doses of Tetanus Toxoid injections), diagnosis of hypertension (blood pressure measurement), and diagnosis of asymptomatic infections (urine testing, blood tests, and HIV test) [11]. The services provided to pregnant women with ANC can reduce the chances of poor pregnancy outcomes such as stillbirths, premature deliveries, and low birth weight [12, 13]. When compared to any single element/component of ANC, compliance with recommended contents of care provided higher protection against preterm birth and low birth weight [14, 15]. Poor quality and inadequate utilization of ANC services throughout pregnancy, on the other hand, may raise the risk of preventable detrimental pregnancy outcomes [14].

Ethiopia implemented the WHO organization (WHO) focused ANC (FANC) model in all health institutions until February 2022, which was a goal-oriented approach to delivering evidence-based interventions at four crucial points throughout pregnancy [16]. However, in 2019, only 62.9% and 43% of pregnant women had at least one and four ANC contacts, respectively [17]. However, in Ethiopia, relatively little attention has been paid to the content of the care provided.

Although numerous studies have been conducted on the uptake of ANC in Ethiopia, practically all have focused on the timing and number of visits [18–20], and the level of adequacy of

services received during visits with their spatial variability has not been thoroughly assessed. Hence, the purpose of this study was to uncover the magnitude, geographic variations, and individual- and community-level determinants of the missing vital elements of ANC in Ethiopia. By identifying specific geographic places with a high magnitude of missing contents of care, resources, and interventions could be prioritized and allocated for those locations needing the most. Furthermore, the identified individual- and community-level barriers to compliance with recommended items of care would be used to improve efforts by various stakeholders such as program planners, policymakers, and healthcare providers to meet Sustainable Development Goal 3(SDG3) [21]. Working on safer pregnancies and timely access to healthcare services could significantly improve maternal and newborn health outcomes.

## Methods

### Data source and population

This study was conducted using the 2016 Ethiopian Demographic and Health Survey (EDHS 2016), which was conducted from January 18, 2016, to June 27, 2016 [22]. Study populations were all women aged 15–49 years who gave birth within the five years before the survey. Women who had complete information on key WHO-recommended components of care during their last ANC visits were included in the study unless excluded.

### Sampling procedure and sample size

A stratified two-stage cluster sampling approach was used to select study participants. The first stage involved selecting 645 enumeration areas (EAs) with a probability proportional to EA size. The second stage involved selecting a fixed number of 28 households per cluster using a systematic equal probability selection of eligible women. Finally, the analysis was based on the records of 4712(weighted sample size of 4771) (Fig 1). Detailed methodologies are addressed in the 2016 EDHS report [22]. The position data of each surveyed cluster (geographic coordinates) were obtained using Global Positioning System (GPS) data. The GPS readings for each cluster were obtained from the center. To preserve respondents' privacy, all survey groups' GPS latitude/longitude positions were randomly displaced. The maximum displacement was two kilometres (km) for urban clusters and five kilometres (km) for 99% of rural clusters. The remaining 1% of rural clusters shifted up to a ten-kilometer distance [23].

### Variables of the study

**Dependent variable.** Seven nationally recommended contents of care during ANC in Ethiopia were assessed to define non-compliance with or missing WHO-recommended components of care. The items included receipt of counselling on danger signs(pregnancy complications), iron supplements, at least two doses of Tetanus Toxoid injections, blood pressure measurement, urine testing, blood tests, and HIV tests as part of ANC [9, 20, 22, 24]. The answers were coded as (1 = Yes, and 0 = No). Based on the responses, a composite index was created with minimum and maximum values of 0 and 7, respectively. Finally, this count variable was dichotomized as missed care (= 1) and full care (= 0).

**Explanatory variables.** After reviewing relevant and current literature, both individual- and community-level factors were considered. Individual-level factors include sociodemographic, obstetric, and healthcare-related characteristics. Community-level factors, were features shared by all women living in the same community (cluster), such as place of residence and region (Table 1).

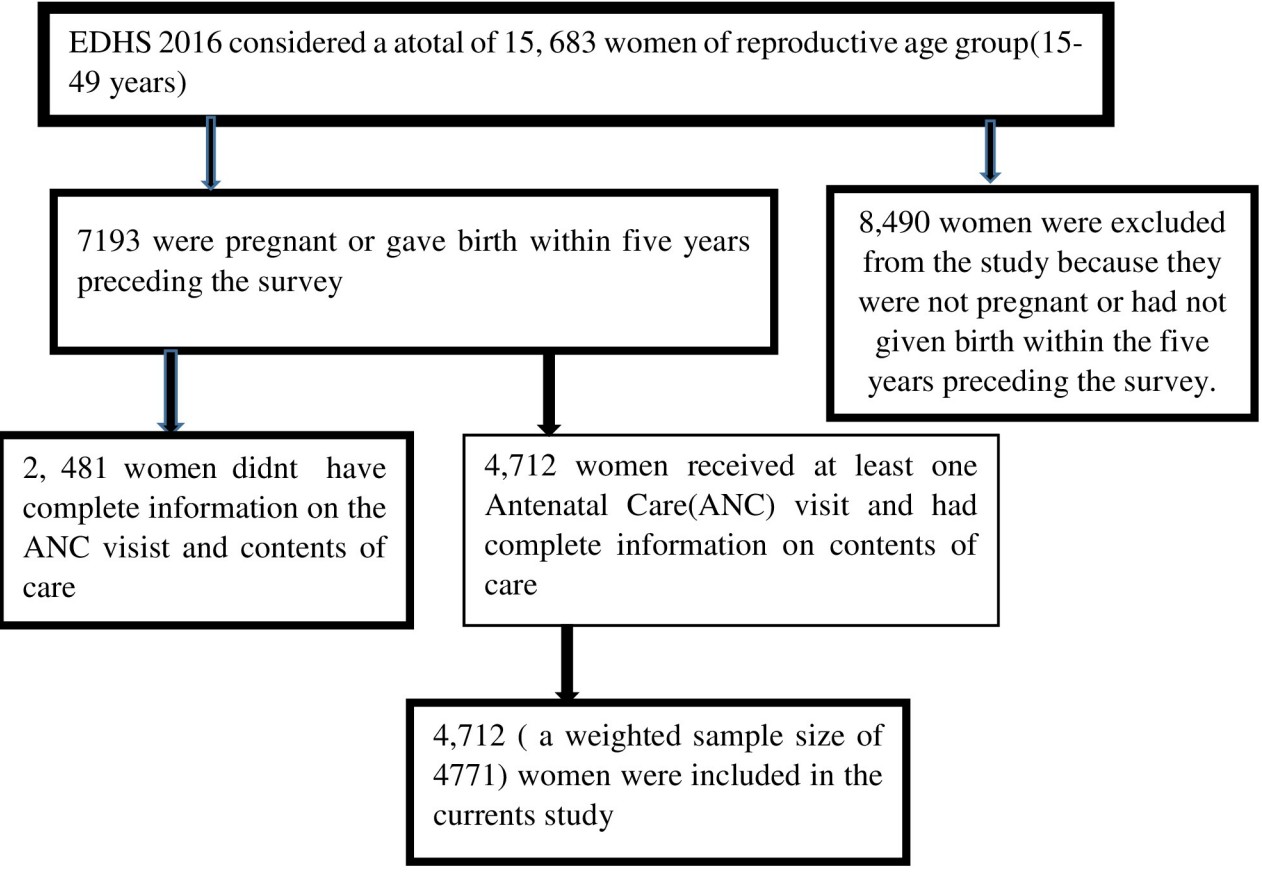

**Fig 1. Schematic presentation of the sampling procedure and sample size determination for the study.**

## Data management and statistical analysis

STATA version 16.1 was used for data cleaning, coding, and analysis, whereas ArcGIS 10.7 and SaTScan were employed for geographical analysis. All data were weighted using a weighting factor $\left(\frac{v005}{1000000}\right)$ to minimize under or over-representation. The weighted proportion of the outcome variable computed in STATA was then generated in Microsoft Excel 2016 (CSV format) and imported into ArcGIS 10.7 for spatial analysis.

**Spatial analysis.** The CSV file containing the outcome variable was first imported into ArcGIS version 10.7 and merged with the GPS data (shape file) to obtain latitude and longitude data. The event data were then transformed into a shape file and displayed on an XY plane. Before initiating the study, geographically coordinated data were projected using the UTS system.

**Spatial autocorrelation (Global Moran's I).** Spatial autocorrelation (Moran's index) was used to determine whether the missing contents of care in Ethiopia were dispersed, clustered, or randomly distributed. This analysis produced a single output value, which ranged from −1 to 1. A Moran's I value close to 1 suggests a significant positive spatial autocorrelation (the outcome of interest is clustered), whereas a value close to -1 indicates a large negative spatial autocorrelation (the outcome is dispersed) across clusters. A Moran's I value near zero implies that the spatial distribution is random [25]. Furthermore, a statistically significant Moran I value (p<0.05) led to rejection of the null hypothesis (missing content care was randomly distributed) [25, 26].

**Table 1. Potential predictors of missing key elements of ANC at individual and community level in Ethiopia, EDHS 2016.**

| Variables | Description |
|---|---|
| Age | The respondent's age, expressed in years, at the time of the survey and categorized as 15–19, 20–34, and 35–49 years |
| Marital status | Categorized as in marital relationships or not in a marital relationship |
| Level of Education | Categorized as no education, primary and secondary/higher education |
| Family size | Number of household members at the time of data collection and categorized as ≤5 or >5 |
| Wealth index | Calculated using straightforward information on a household's ownership of certain goods, such as televisions and bicycles; housing materials; livestock, crop production, and access to water, sanitation, and hygiene. Finally, it was grouped into richest, richer, middle, poorer, and poorest |
| Parity | The number of living children the woman had at the time of the survey and grouped into Primiparous, Multiparous, and Grand multiparous |
| Antenatal care | Number of women who received antenatal care for their last birth, which was initially reported in continuous form and then grouped as no antenatal care, 1 visit, 2–3 visits, 4+ visits |
| Timing ANC | Numbers of women who received ANC for their last birth according to the single number of months they were pregnant at the time of the first visit, and grouped as 1st, 2nd, and 3rd trimester |
| Pregnancy status during last childbirth | Categorized unwanted (wanted later and not wanted at all) and wanted (wanted then) |
| Media exposure | The number of women exposed to specific media at various frequencies, such as reading a newspaper, watching television, and listening to the radio. Categorized as: Not at all, Less than once a week, and At least once a week |
| Autonomy in decision-making | The total number of married women between the ages of 15 and 49 who make decisions for their health care, major household buys, and visits to family or relatives. |
| **Community level factors** | |
| Residence | The area where respondents lived when the survey was conducted and grouped as Urban and Rural |
| Region | The geographically delineated area where the woman was resided at the time of the survey. Three categories were created as Small periphery regions (Afar, Somalia, Benishangul, and Gambella), Major central regions (SNNPRs, Tigray, Amhara, and Oromia), and Metropolitans (Addis Ababa, Dire Dawa, and the Harari region) |

**Hot spot analysis (Getis-Ord Gi\* statistics).** Getis-Ord Gi\* statistics were run to detect significant hot spots and cold spots for missing key elements of the ANC. To determine the statistical significance of clustering, the Z-score was obtained and the p-value was determined. A statistical output with a high GI suggests a "hot spot," whereas one with a low GI indicates a "cold spot" [27, 28].

**Spatial interpolation.** Using data from the sampled EAs, spatial interpolation was performed using the Ordinary Kriging technique to forecast the status of missing care contents in unsampled areas. The ordinary Kriging spatial interpolation technique was chosen because it has numerous advantages over other techniques. First, it considers the spatial dependence or autocorrelation in the data, which is vital to effectively predicting unobserved values based on values observed at nearby places [29]. It assumes that the statistical features of the data, such as the mean and variance, are constant across the research area, which is known as the assumption of stationarity, which simplifies the modelling process and often works well in practice [30]. In addition, it reduces prediction variance by providing more weight to nearby sample points with comparable values, which aids in producing more reliable estimates, particularly in areas where data is sparse [31]. Furthermore, because ordinary Kriging is based on

geostatistical models that account for both spatial trend and spatial variability in data, it provides a more accurate depiction of the underlying spatial patterns in ANC content [30].

**Spatial scan statistical analysis.**   Sat Scan analysis was performed to identify substantial hot spots of missing ANC services using Kuldorff's SaTScan version 9.6 statistical software. The Bernoulli-based approach was used to find statistically significant spatial clusters with a high number of women who missed the content of care [32]. The main reasons for using Bernoulli model is because it accommodates binary outcomes; the occurrence of women missing the content of antenatal care (coded as 1 = cases), and the non-occurrence would be women receiving the recommended antenatal care contents (coded as 0 = controls). In addition, the results of the Bernoulli-based spatial scan analysis tend to be flexible and easy to interpret, such as revealing clusters with high or lower risk factors, making it easier for policymakers and public health experts to understand and focus interventions [33, 34]. The SaTScan statistics gradually scanned the area to determine the number of observed and expected events inside the window at every point. The default maximum spatial cluster size of 50% of the population was set as an upper limit, allowing both small and large clusters to be recognized. The scanning window with the highest likelihood was selected as the most likely and high-performing cluster to represent a case, and the level of significance was set at p-value<0.05 [32, 35].

**Multilevel mixed-effect logistic regression.**   Given the hierarchical nature of the EDHS data, multilevel modelling was employed. Because the outcome variable was in binary form, we used multilevel multivariable regression analysis.

**Model building and selection.**   *Fixed effects (measures of association)*. A multilevel bivariable logistic regression analysis was first carried out to examine the influence of each predictor on the response variable, and variables with a p-value< 0.25 were accounted for in the multilevel multivariable logistic regression. The Variance Inflation Factor (VIF) was used to look for multicollinearity across covariates at a cut-off point of 10 and none was found (the VIF ranged from 1.02 to 2.33 with a mean of 1.44). Finally, multilevel multivariable logistic regression analysis was performed to identify significant predictors of missing contents of care, and statistically significant variables (p <0.05) were reported with their corresponding 95% confidence intervals.

*Random effects*. Four different models were fitted by using multilevel methods. Model one (null model) had no explanatory variable. The second and third models contained only individual- and community-level factors, respectively. The fourth (complete) model incorporated all individual- and community-level characteristics. The intraclass correlation coefficient (ICC) and proportional change in variance (PCV) were used to quantify the random effects in each model (variability in the level of missing care contents between and across clusters).

$$ICC = \frac{var(b)}{Var(b) + Var(w)},$$

where Var(b) is the variance at the group level and Var(w) is a predicted individual variance component, which is $\pi^2/3 \approx 3.29$.

Proportional Change in Variance (PCV) was estimated as

$$PCV = \frac{(Va - Vb)}{Va} * 100,$$

where $V_a$ is the variance of the initial model (null model) and $V_b$ = variance of the subsequent models (models 2, 3, 4, and 5).

*Goodness of fit*. Deviance = (-2 * (Log Likelihood (LL) of each model), Schwarz's Bayesian Information Criterion (BIC), and Akaike's information criterion (AIC) were used to assess

goodness of fit. After comparison, the fourth model with the lowest deviation, AIC, and BIC values was selected as the best-fit model for this study (Table 6).

## Ethical consideration and consent to participate

All methods and procedures were performed according to the relevant guidelines and regulations of the Declaration of Helsinki. The DHS office provided us with written permission to use both the DHS and GPS datasets following registration. Furthermore, as this is secondary data, the ethics committee of Wachemo University College of Medicine and Health Sciences declared that no formal ethics approval was required with a written letter at the reference number (WCU/327/2023).

## Results

### Sociodemographic characteristics of the respondents

The current study examined a total weighted sample of 4771 women, with the mean (±SD) age 28.79(±6.61) years. Most participants (90.4%) were from the central region (Oromia, Amhara, and Tigrai). Rural residents accounted for 81.7% of the study participants. Orthodox and Muslim religious followers accounted for 42.5% and 33.0% of respondents, respectively. The vast majority (88.2%) of women missed at least one key element of care during their prenatal visit (Table 2). Nine in every ten women from the poorest wealth quintile (93.9%) and without formal education (91.9%) did not receive full content of care. The highest mean number of contents of care was observed among women who lived in urban areas (5.46) or with the richest wealth quintile (5.71) (Table 2).

### Obstetric and health service-related- characteristics

Nearly half (47.0%) of the participants were multiparous. In terms of time and frequency of ANC visits, nearly one-third (32.5%) of the women had their first visit during the first trimester, and half (50.6%) had four or more visits. Only 5.0% of the women were enrolled in any health insurance scheme. Nearly three-quarters, 74.0% of the women were autonomous in decision-making. Women who commenced ANC in the third trimester and received only one ANC visit failed to receive full content of care in 97.7% and 97.0% of cases, respectively (Table 3).

### The level of missing key contents of ANC

According to the analysis, 88.2% of the women failed to receive full care during their last prenatal visit. The Somali region had the largest proportion of women who missed full contents of care during prenatal visits (94.9%), closely followed by the Oromia region (93.7%). The top three types of care that the majority of women did not receive as part of ANC were HIV tests (59.0%), information about pregnancy complications (55.0%), and receipt of at least two doses of TT vaccination (43.1%) (Table 4).

### Results of spatial analysis

**Spatial distribution of missing ANC contents.** Western Oromia, northeast of SNNPR, western border of Afar, and southern border of Benishangul Gumuz were identified as regions with a high proportion of women who missed ANC content. In contrast, Tigray, Addis Ababa, and Dire Dawa had lower proportions (Fig 2).

The global spatial autocorrelation analysis found that the geographical distribution of missing contents of care across the country was non-random (i.e., there was significant spatial

**Table 2. The distribution of missing key contents of care during prenatal visits across sociodemographic characteristics of women in Ethiopia, EDHS 2016.**

| Variable categories | Total [Weighted frequency (%)] | Missing ANC contents | | Mean number of ANC Contents received | |
|---|---|---|---|---|---|
| | | Yes [n(%)] | cOR (95% CI) | Mean (95% CI) | p-value |
| **Current age of women** | | | | | |
| 15–24 | 1,236(25.9) | 1,069(86.5) | Ref. | 4.71(4.59, 4.78) | 0.921[a] |
| 25–34 | 2,496(52.3) | 2,198(88.0) | 1.35(0.97, 1.89)* | 4.64(4.57, 4.71) | |
| 35–49 | 1,039(21.8) | 940(90.5) | 1.96(1.27, 3.02)* | 4.41 (4.30, 4.52) | |
| **Regions** | | | | | |
| Major central regions | 4,311(90.4) | 3,823(88.7) | 2.96(2.11, 4.15)* | 4.45 (4.38, 4.53) | <0.001[a] |
| Peripheral | 226(4.7) | 209(92.9) | 4.95(3.31, 7.38)* | 4.3(4.20, 4.39) | |
| Metropolitans | 234(4.9) | 174(74.4) | Ref. | 5.36 (5.28, 5.45) | |
| **Religion** | | | | | |
| Orthodox | 2,030(42.5) | 1,687(83.1) | Ref. | 5.01(4.94, 5.08) | 0.052[a] |
| Muslim | 1,573(33.0) | 1,454(92.5) | 2.34(1.60, 3.43)* | 4.42 (4.34, 4.50) | |
| Protestant | 1,049(22.0) | 952(90.8) | 1.39(0.86, 2.23)* | 4.16(4.04, 4.29) | |
| Others | 119(2.5) | 113(94.5) | 1.83(0.74, 4.54)* | 4.15(3.67, 4.63) | |
| **Residence** | | | | | |
| Urban | 875(18.3) | 678(77.5) | Ref. | 5.46(5.38, 5.51) | <0.001[b] |
| Rural | 3,896(81.7) | 3,528(90.6) | 3.56(2.63, 4.81)* | 4.25(4.18, 4.31) | |
| **Wealth index combined** | | | | | |
| Poorest | 794(16.7) | 746(93.9) | 4.84(3.22, 7.27)* | 3.90(3.83, 4.05) | <0.001[a] |
| Poorer | 935(19.6) | 871(93.1) | 2.98(2.00, 4.44)* | 4.22(4.09, 4.35) | |
| Middle | 996(20.9) | 912(91.6) | 2.52(1.69, 3.74)* | 4.24(4.11, 4.38) | |
| Richer | 967(20.2) | 845(87.4) | 1.62(1.13, 2.33)* | 4.57(4.44, 4.71) | |
| Richest | 1,078(22.6) | 832(77.1) | Ref. | 5.71(5.48, 5.82) | |
| **Educational status** | | | | | |
| No education | 2,580(54.1) | 2372(91.9) | 5.30(3.47, 8.06)* | 4.17(4.09, 4.24) | <0.001[a] |
| Primary | 1,576(33.0) | 1396(88.5) | 3.12(2.01, 4.85)* | 4.77(4.69, 4.85) | |
| Secondary and higher | 614(12.9) | 438(71.3) | Ref. | 5.53(5.43, 5.62) | |

Key: Ref: Reference Category, cOR: crude odds ratio,

*: significant at p-value<0.25.

[a]p-values were based on an independent t-test

[b]p-values were based on analysis of variance(ANOVA)

variation) (global Moran's I = 0.39, p<0.001, z-score of 12.96). The clustered patterns (on the right side) indicate that missing WHO-recommended items of care occurred frequently throughout the country (Fig 3).

**Hot spot (Getis-Ord Gi\*) analysis of missing ANC contents.** In the hot spot analysis, significant spatial clusters (Hot Spots) for incomplete contents of care were detected in the eastern and northern areas of the SNNPR, as well as in the western border of the Oromia regions. Eastern Tigray, the central part of Addis Ababa, Dire Dawa, and Harari, on the other hand, had a small proportion of women with incomplete contents of care (Cold Spots) (Fig 4).

**Spatial interpolation.** The ordinary Kriging technique was used to estimate the spatial distribution of missing full contents of care for areas where data were not collected. Accordingly, the southern portions of Oromia, the southern and eastern parts of SNNPR, the western parts of Addis Ababa, and the northern part of Afar had the highest expected prevalence of missing the recommended contents of care (red shaded). In contrast, the projected proportion

**Table 3. The distribution of missing key contents of care during prenatal visits across obstetric and health service-related- characteristics of women in Ethiopia, EDHS 2016.**

| Variable categories | Total [Weighted frequency (%)] | Women who missed contents of care | | Mean number of ANC contents received | |
|---|---|---|---|---|---|
| | | Yes [n(%)] | cOR(95% CI) | Mean (95% CI) | p-value |
| **Parity** | | | | | |
| Primiparous | 1,193(25.0) | 1,033(84.8) | Ref. | 4.95(4.86, 5.04) | 0.762[a] |
| Multiparous | 2,241(47.0) | 1,961(87.5) | 1.29(0.92, 1.79) | 4.64(4.57, 4.71) | |
| Grand multiparous | 1,307(27.4) | 1,21(92.8) | 2.14(1.37, 3.34)* | 4.17(4.06, 4.27) | |
| **Frequency of ANC** | | | | | |
| One visit | 349(7.3) | 338.8(97.0) | 7.23(3.32, 15.75)* | 3.39(3.19, 3.60) | <0.001[b] |
| 2–3 visits | 2,007(42.1) | 1,869(93.1) | 2.59(1.85, 3.64)* | 4.17(4.09, 4.25) | |
| ≥4 visits | 2,414(50.6) | 1,998(82.8) | Ref. | 5.07(5.00, 5.12) | |
| **Timing of ANC** | | | | | |
| 1st Trimester | 1,550(32.5) | 1,289(83.2) | Ref. | 5.03(4.95, 5.10) | <0.001[b] |
| 2nd Trimester | 3,006(63.0) | 2,707(90.0) | 1.65 (1.22, 2.22)* | 4.41(4.34, 4.48) | |
| 3rd Trimester | 215(4.5) | 210(97.7) | 7.51(2.05, 17.56)* | 3.12(2.84, 3.40) | |
| **Planning status of pregnancy** | | | | | |
| Wanted | 3,622(75.9) | 3,176(87.7) | 1.46(0.98, 2.16) | 4.61(4.55, 4.67) | 0.472[b] |
| Unwanted | 1,149(24.1) | 1,030(89.9) | Ref. | 4.57(4.46, 4.68) | |
| **Listen to radio** | | | | | |
| Not at all | 3,162(66.3) | 2,843(89.9) | 1.53(1.04, 2.26)* | 4.40(4.34, 4.47) | 0.001[a] |
| Less than once a week | 793(16.6) | 675(85.1) | 1.06(0.69, 1.63) | 4.93(4.81, 5.05) | |
| At least once a week | 817(17.1) | 688(84.2) | Ref. | 5.10(4.99, 5.22) | |
| **Watching TV** | | | | | |
| Not at all | 3,564(74.7) | 3,249(91.2) | 3.84(2.66, 5.53)* | 4.20(4.18, 4.31) | <0.001[a] |
| Less than once a week | 552(11.6) | 472(85.7) | 2.18(1.32, 3.59)* | 5.03(4.88, 5.17) | |
| At least once a week | 656(13.7) | 484(73.8) | Ref. | 5.54(5.47, 5.62) | |
| **Reading newspaper** | | | | | |
| Not at all | 4,295(90.0) | 3,846(89.6) | 1.78(0.82, 3.88) | 4.86(4.41, 4.52) | 0.091[a] |
| Less than once a week | 353(7.4) | 261(74.0) | 0.62(0.25, 1.51) | 5.60(5.47, 5.74) | |
| At least once a week | 123(2.6) | 99(80.2) | Ref. | 5.63(5.43, 5.84) | |
| **Own mobile phone** | | | | | |
| Yes | 1,166(24.4) | 908(77.9) | Ref. | 5.32(5.25, 5.39) | <0.001[b] |
| No | 3,605(75.6) | 3,298(91.5) | 3.38(2.38, 4.80) * | 4.22(4.16, 4.29) | |
| **Covered by Health Insurance** | | | | | |
| Yes | 241(5.0) | 203(84.3) | Ref. | 5.04(4.83, 5.25) | 0.799[b] |
| No | 4,530(95.0) | 4,003(88.4) | 1.13(0.65, 1.96) | 4.58(4.53, 4.63) | |
| **Autonomy in decision-making** | | | | | |
| Autonomous | 3,530(74.0) | 3,087(87.5) | Ref. | 4.69(4.64, 4.75) | 0.031[b] |
| Non-autonomous | 1,241(26.0) | 1,119(90.2) | 1.42(1.02, 1.97)* | 4.35(4.25, 4.45) | |
| **Ease of distance to seek medical care** | | | | | |
| Big problem | 2,401(50.3) | 2,193(91.3) | 1.42(1.01, 2.00)* | 4.19(4.11, 4.27) | <0.001[b] |
| Not a big problem | 2,370(49.7) | 2,013(85.0) | Ref. | 4.93(4.87, 5.00) | |
| **Access to money for seeking medical care** | | | | | |
| Big problem | 2,560(53.7) | 2,340(91.1) | 1.66(1.20, 2.29)* | 4.32(4.25, 4.40) | 0.011[b] |
| Not a big problem | 2,211(46.3) | 1,867(84.4) | Ref. | 4.88(4.81, 4.95) | |

Key: Ref: Reference Category, cOR: crude odds ratio,

*: significant at p-value<0.25.

[a] p-values were based on an independent t-test,

[b] p-values were based on analysis of variance(ANOVA)

**Table 4. The level of missing ANC contents across regions of Ethiopia, EDHS 2016.**

| Regions | Total | Women who missed each care [Weighted frequency (%)] | | | | | | |
|---|---|---|---|---|---|---|---|---|
| | | Told about pregnancy complications | BP measurement | Taking Blood sample | Taking urine sample | Iron supplementation | HIV test as part of ANC | TT vaccination |
| Tigray | 486(10.2) | 223(45.9) | 50(10.4) | 45(9.3) | 86(17.8) | 76(15.6) | 212(43.7) | 267(55.0) |
| Afar | 37(0.8) | 24(66.5) | 10(26.5) | 9(23.8) | 10(28.3) | 12(32.7) | 22(59.1) | 17(47.5) |
| Amhara | 1104(23.1) | 583(52.9) | 251(22.7) | 219(19.8) | 342(31.0) | 292(26.4) | 571(51.8) | 589(53.3) |
| Oromia | 1607(33.7) | 1045(62.0) | 489(30.4) | 605(37.6) | 635(39.0) | 869(54.0) | 1151(71.6) | 579(36.0) |
| Somali | 118(2.5) | 82(69.3) | 21(18.0) | 30(25.4) | 32(27.0) | 53(44.9) | 90(76.8) | 39(33.2) |
| Benishangul | 56(1.2) | 28(49.4) | 19(34.6) | 15(27.1) | 20(36.7) | 19(33.4) | 38(67.3) | 22(39.9) |
| SNNPR | 1114(23.4) | 558(50.0) | 326(29.2) | 379(34.0) | 482(43.2) | 494(44.3) | 733(65.8) | 456(40.9) |
| Gambella | 15(0.3) | 9(61.9) | 4(24.1) | 3(17.2) | 1(0.7) | 7(45.0) | 8(54.6) | 6(37.7) |
| Harari | 13(0.3) | 6(47.1) | 1(9.9) | 2(12.2) | 2(14.0) | 5(36.2) | 7(53.8) | 3(19.5) |
| Addis Ababa | 192(6.7) | 48(25.1) | 5(2.7) | 2(1.3) | 1(0.8) | 67(35.4) | 69(35.9) | 71(36.9) |
| Diredawa | 29(0.6) | 19(65.7) | 3(11.7) | 2(8.2) | 3(8.7) | 10(33.3) | 15(50.3) | 9(29.7) |
| **Total** | 4,771 (100%) | 2,626 (55.0) | 1180 (24.7) | 1,310 (27.5) | 1,617 (33.9) | 1904 (39.9) | 2,816 (59.0) | 2058 (43.1) |

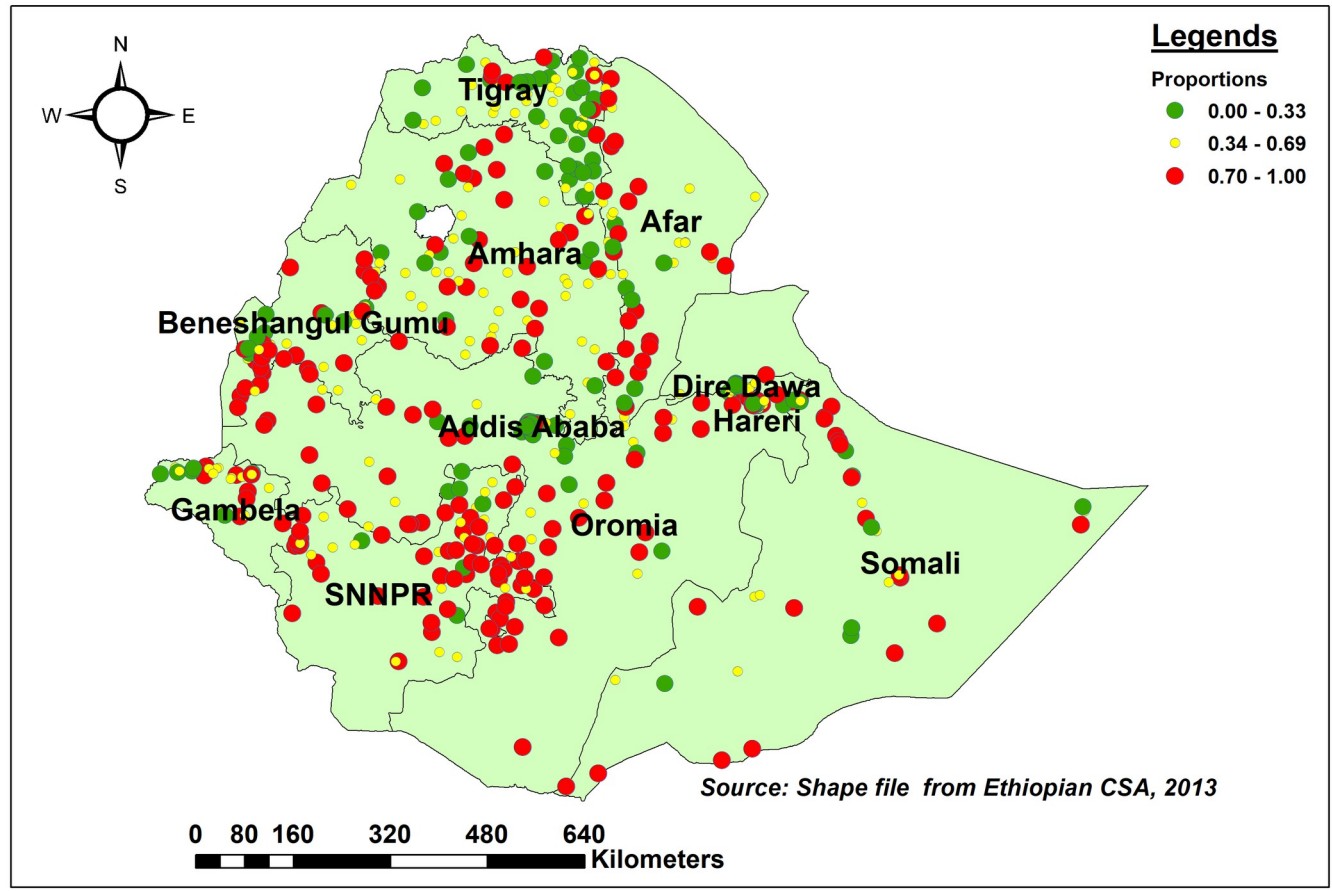

**Fig 2. Spatial distribution of missing ANC contents in Ethiopia, EDHS 2016.**

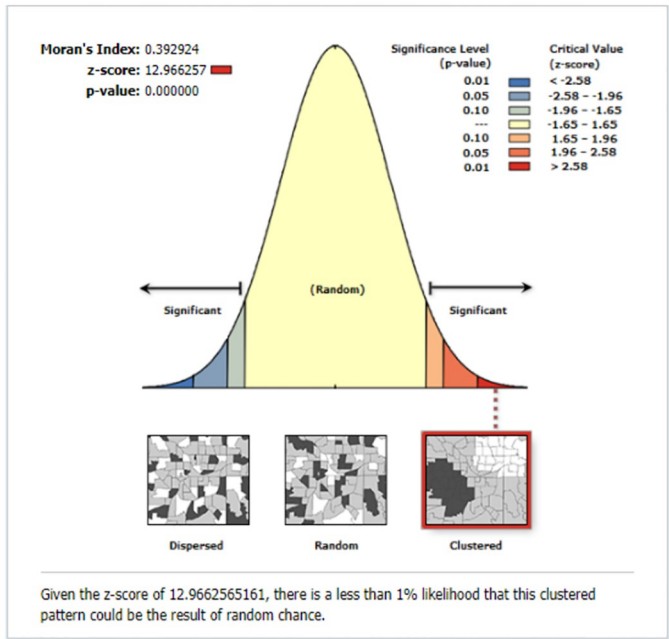

**Fig 3. The global spatial autocorrelation of missing ANC contents in Ethiopia, EDHS 2016.**

of women with incomplete care items (green shaded) was low in Tigray and Central Addis Ababa (Fig 5).

**Spatial scan statistical (SaTScan) analysis.** The SaTScan spatial analysis showed two statistically significant groups of SaTScan clusters, with a high number of women who missed key contents of care. In addition, the analysis revealed 156 significant clusters that accounted for missing contents of care, with 133 and 23 found in the first and second most likely clusters, respectively. The first most likely cluster is located at geographical coordinates of (5.546952 N, 37.666334 E) with a 379.77 Km radius, and LLR of 85.76 at p<0.001. Women living in these areas had a 45% (RR = 1.45) higher risk of missing ANC content than those living elsewhere (Table 5).

## Results of multilevel mixed effect logistic regression

**Random effect (measures of variation).** For the measures of variation (random effects), the intraclass correlation coefficient (ICC) and Proportional Change in Variance (PCV) were computed. The results of the null model (Model I) revealed that variability between clusters accounted for 17.1% of the total variation in missing ANC content (ICC = 0.171, p<0.001). Furthermore, individual and community variability accounted for 8.1% (ICC = 0.081, p<0.001) and 11.6% (ICC = 0.116, p<0.001) of the variation in the missing care content, respectively. Individual- and community-level factors together accounted for 58.5% of the national variation observed in the null model (PCV = 58.5%). As we moved from Model I (null model) to Model IV (full model), the values of AIC, BIC, and Deviance decreased, indicating that the final model was best fitted (Table 6).

**Predictors of missing contents of ANC.** The factors that have a significant association with missing ANC contents were educational status, frequency and timing of ANC, religion, ownership of mobile phone, and residence (Table 7).

## Hot Spot and Cold Spot for receiving incomplete contents care during ANC in Ethiopia

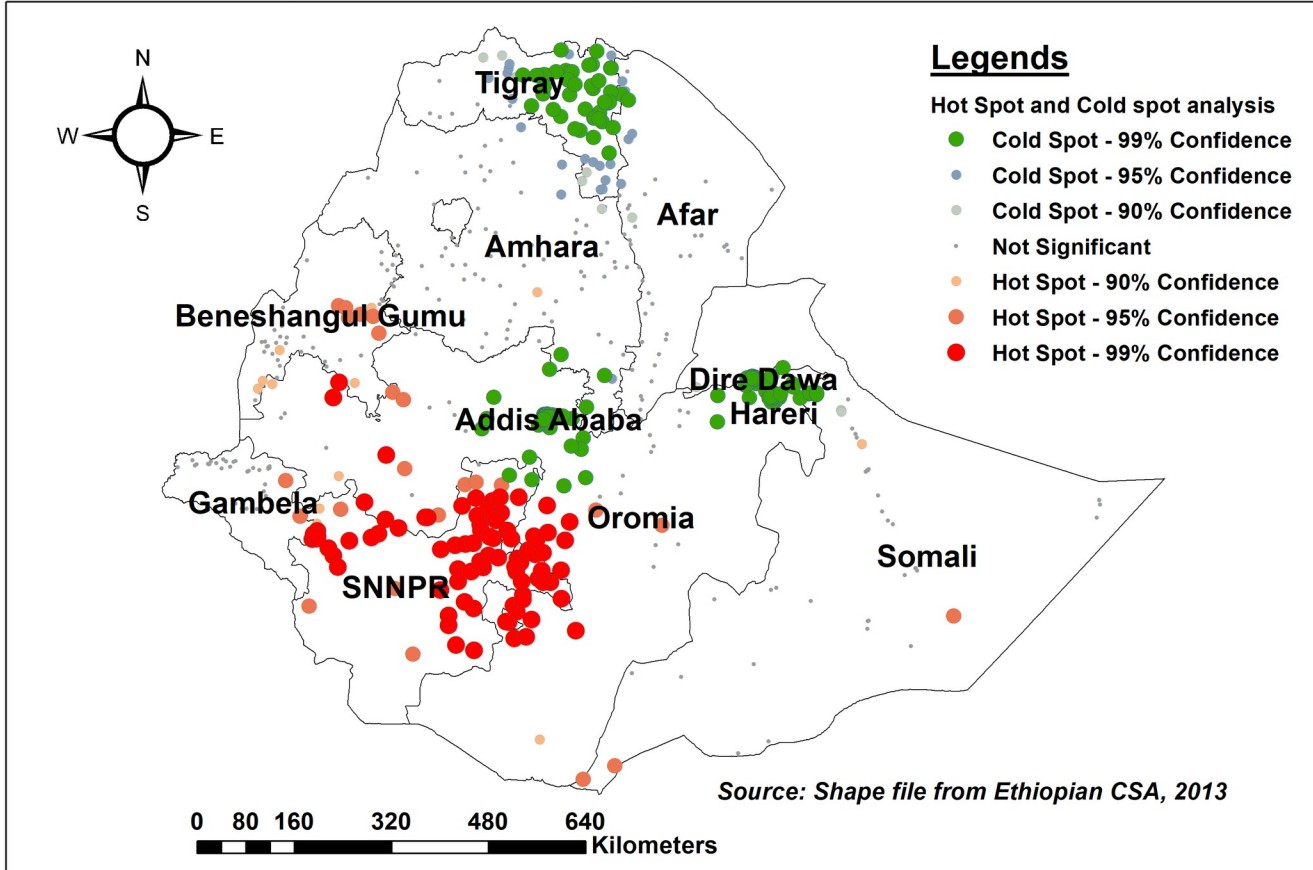

**Fig 4. Hot spot and cold spot analysis of missing ANC contents across regions in Ethiopia, EDHS 2016.**

Women without formal education were two times more likely to miss the contents of ANC than women who attended secondary and college education. The odds of missing contents of ANC were 1.94 times higher among women without formal education than among women who attended secondary and college education (aOR = 1.94, 95% CI: 1.24, 3.02). The timing of ANC was identified as significant predictors of missing ANC content. Women who initiated their first ANC visit in the third trimester were 3.05 times more likely to miss ANC content (aOR = 3.05, 95% CI: 1.59, 6.54) than women who began in the first trimester. Similarly, the odds of missing content of care were 4.13 times higher among women who received only one visit than among those who received four or more visits (aOR = 4.13, 95% CI:1.95, 8.74). Women who did not own a mobile phone were 1.44 times more likely to miss the contents of ANC than those who did (aOR = 1.44, 95% CI: 1.07, 1.95). Finally, women who resided in the rural part of the country had a 1.68 times higher chance of missing key contents ANC compared to their urban counterparts (aOR = 1.68, 95% CI:1.47, 2.71) (Table 7).

## Discussion

This study assessed the magnitude and determinants of non-compliance with the WHO-recommended contents of ANC in Ethiopia. The vast majority (88.2%) of pregnant women failed

Interpolated spatial distribution of women with incomplete contents of care during prenatl visist in Ethiopia

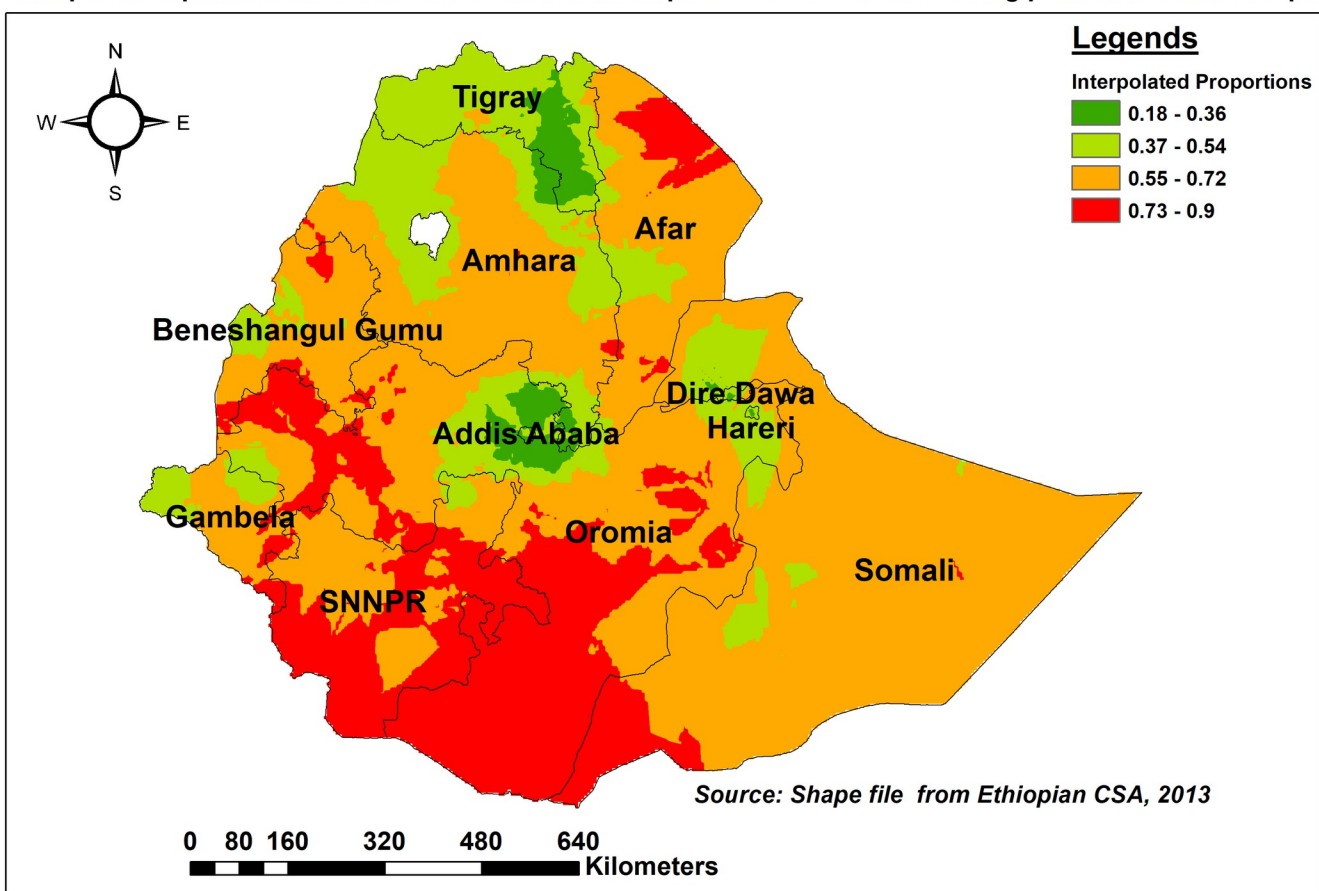

**Fig 5. Ordinary Kriging interpolation of the spatial distribution of missing ANC contents in Ethiopia, EDHS 2016.**

to receive full contents of care during prenatal visit. There was significant spatial variation in the missing content of care across regions. Educational status, timing and adequacy of ANC visits, ownership of mobile phones, and residence were identified as significant predictors of missing full contents of ANC among respondents.

**Table 5. The most likely SaTScan clusters of areas with a high proportion of women who missed full contents of ANC in Ethiopia, EDHS 2016.**

| Most likely clusters | Enumeration areas (clusters) identified | Number of clusters | Population | No. of case | Coordinates / Radius | Relative risk | LLR | P-Value |
|---|---|---|---|---|---|---|---|---|
| 1st most likely cluster | 50, 342, 86, 21, 503, 450, 574, 182, 505, 398, 232, 87, 32, 466, 316, 406, 600, 434, 445, 180, 141, 20, 634, 215, 53, 408, 347, 162, 313, 468, 216, 470, 34, 148, 405, 422, 576, 306, 126, 113, 565, 601, 373, 12, 308, 388, 518, 609, 365, 360, 420, 41, 391, 578, 537, 331, 589, 227, 297, 272, 447, 486, 26, 76, 502, 154, 223, 207, 577, 271, 489, 376, 377, 14, 619, 522, 633, 477, 142, 432, 359, 174, 325, 82, 337, 437, 204, 54, 7, 586, 562, 262, 438, 371, 168, 552, 465, 177, 459, 243, 62, 197, 299, 526, 289, 213, 554, 217, 46, 524, 394, 119, 472, 411, 123, 326, 118, 555, 23, 452, 290, 149, 319 | 123 | 1023 | 743 | (5.546952 N, 37.666334 E) / 379.77 km | 1.45 | 85.76 | <0.001 |
| 2nd most likely cluster | 349, 70, 304, 621, 88, 124, 320, 65, 335, 161, 569, 17, 374, 175, 184, 416, 244, 563, 294, 462, 558, 433, 248, 209, 183, 6, 150, 595, 364, 407, 137, 395, 35 | 33 | 293 | 211 | (9.528826 N, 35.631217 E) / 126.51 km | 1.34 | 19.11 | <0.001 |

**Table 6. Random intercept variances and model fit statistics comparison of multilevel mixed effect logistic regression model.**

| Measures | Model I (null model) | Model II (individual-level factors) | Model III (community-level factors) | Model-IV (full model) |
|---|---|---|---|---|
| **Random effects** | | | | |
| Variance | 0.68 | 0.287 | 0.434 | 0.282 |
| ICC | 0.171 | 0.081 | 0.116 | 0.079 |
| AIC | 3590.4 | 3336.6 | 3494.0 | 3337.2 |
| BIC | 3603.3 | 3504.5 | 3526.3 | 3518.0 |
| PCV | Reference | 57.8% | 36.2% | 58.5% |
| **Model fitness** | | | | |
| Log-likelihood | -1793.2 | -1642.3 | -1742.0 | -1640.6 |
| Deviance | 3586.4 | 3284.6 | 3484.0 | 3281.2 |

**Table 7. Results of a multilevel multivariable logistic regression to identify the determinants of missing key contents of ANC in Ethiopia, EDHS 2016.**

| Variable categories | Model I (null model) | Model II (individual-level factors) | Model III (community-level factors) | Model-IV (full model) |
|---|---|---|---|---|
| | | aOR(95%CI) | aOR (95%CI) | aOR (95%CI) |
| **Current age of women** | | | | |
| 15–24 | | Ref | | Ref |
| 25–34 | | 1.33(0.94, 1.87) | | 1.01(0.77, 1.32) |
| 35–49 | ' | 1.64(1.04, 2.59)* | | 1.07(0.76, 1.50) |
| **Religion** | | | | |
| Orthodox | | Ref | | Ref |
| Muslim | | 1.72(1.18, 2.50)* | | **1.95(1.38, 2.76)*** |
| Protestant | | 1.24(0.79, 1.95) | | **1.53(1.13, 2.06)*** |
| Others | | 1.54(0.27, 8.58) | | 1.35(0.45, 3.98) |
| **Wealth index combined** | | | | |
| Poorest | | 1.82(0.87, 3.83) | | 1.09(0.66, 1.78) |
| Poorer | | 2.09(1.07, 4.11)* | | 1.02(0.63, 1.66) |
| Middle | | 1.63(0.88, 3.01) | | 1.39(0.88, 2.19) |
| Richer | | 1.30(0.72, 2.34) | | 1.11(0.75, 1.83) |
| Richest | | Ref | | Ref |
| **Educational status** | | | | |
| No education | | 2.14(1.45, 3.32)* | | **1.94(1.24, 3.02)*** |
| Primary | | 1.48(1.01, 2.16)* | | 1.41(1.00, 2.16) |
| Secondary and higher | | Ref | | Ref |
| **Frequency of ANC** | | | | |
| One visit | | 4.19(1.96, 8.79)* | | **4.13(1.95, 8.74)*** |
| 2–3 visits | | 2.01(1.38, 2.91)* | | **2.02(1.39, 2.93)*** |
| ≥4 visits | | Ref | | Ref |
| **Timing of ANC** | | | | |
| 1st Trimester | | Ref | | Ref |
| 2nd Trimester | | 1.58(1.14, 1.94)* | | **1.45(1.13, 1.85)*** |
| 3rd Trimester | | 3.26(1.59, 6.65)* | | **3.05(1.59, 6.54)*** |
| **Listen to radio** | | | | |
| Not at all | | 0.89(0.72, 1.09) | | 0.97(0.68, 1.40) |
| Less than once a week | | 0.86(0.68, 1.09) | | 0.86(0.56, 1.30) |
| At least once a week | | Ref | | Ref |
| **Watching TV** | | | | |

*(Continued)*

**Table 7.** (Continued)

| Variable categories | Model I (null model) | Model II (individual-level factors) | Model III (community-level factors) | Model-IV (full model) |
|---|---|---|---|---|
| | | aOR(95%CI) | aOR (95%CI) | aOR (95%CI) |
| Not at all | | 1.21(0.94, 1.56) | | 1.19(0.76, 1.85) |
| Less than once a week | | 1.01(0.76, 1.33) | | 1.06(0.64, 1.76) |
| At least once a week | | Ref | | Ref |
| **Own mobile phone** | | | | |
| Yes | | Ref | | Ref |
| No | | 1.48(0.94, 2.31)* | | **1.44 (1.07, 1.95)*** |
| **Autonomy in decision-making** | | | | |
| Autonomous | | Ref | | Ref |
| Non-autonomous | | 1.17(0.83, 1.64) | | 0.90(0.69, 1.19) |
| **Ease of distance to health facility** | | | | |
| Big problem | | 1.24(1.06, 1.86)* | | 1.13(0.93, 1.71) |
| Not a big problem | | Ref | | Ref |
| **Access to money** | | | | |
| Big problem | | 0.93(0.79, 1.08) | | 0.91(0.71, 1.16) |
| Not a big problem | | Ref | | Ref |
| **Regions** | | | | |
| Major central regions | | | 1.34(1.02, 1.75)* | 0.96(0.66, 1.41) |
| Peripheral | | | 1.83(1.35, 2.47)* | 0.88(0.59, 1.31) |
| Metropolitans | | | Ref | Ref |
| **Residence** | | | | |
| Urban | | | Ref | Ref |
| Rural | | | 2.69(1.99, 3.15)* | **1.68(1.47, 2.71)*** |
| **Random effects** | | | | |
| Variance | 0.68 | 0.287 | 0.434 | 0.282 |
| ICC | 0.171 | 0.081 | 0.116 | 0.079 |
| AIC | 3590.4 | 3336.6 | 3494.0 | 3337.2 |
| BIC | 3603.3 | 3504.5 | 3526.3 | 3518.0 |
| PCV | Ref. | 57.8% | 36.2% | 58.5% |
| **Model fitness** | | | | |
| Log-likelihood | -1793.2 | -1642.3 | -1742.0 | -1640.6 |
| Deviance | 3586.4 | 3284.6 | 3484.0 | 3281.2 |

**Key:** Ref.: Reference category; aOR = Adjusted Odds Ratio,

* statistically significant at p <0.05,

** statistically significant at p <0.001

In this study, the level of missing key items of care was consistent with a study conducted in East Africa (88.2%) [36]. However, our estimate was higher than previous studies in Nepal (77%) [37], Bangladesh (82%) [38], and Rwanda (75.64%) [39]. This disparity could be attributed to differences in socioeconomic and sociocultural characteristics, availability and accessibility of health service infrastructure, and maternal health service coverage between countries. Significant spatial clusters (Hot Spots) for incomplete contents of care were detected in the eastern and northern borders of SNNPR, as well as in the western border of the Oromia regions. These border areas in Ethiopia are known for their low uptake of maternal health services due to fewer healthcare facilities, limited transportation, inadequate healthcare providers, lower socioeconomic conditions, and a lack of medical supplies and equipment to access

maternal health services, making it difficult for pregnant women to access recommended items of care [40, 41].

The odds of missing care content were found to be higher among women with no formal education. This is supported by similar studies conducted in East Africa [36], Nigeria [24], Ghana [42] and Ethiopia [20]. This could be because women with no formal education may have limited access to prenatal care information and may be unaware of the essential elements of care that they should receive. Similarly, these women might encounter difficulties interpreting medical terms, instructions, or advice delivered to them during ANC visits, resulting in missing essential contents of care. Furthermore, these women may encounter financial constraints that limit their access to ANC, which can result in delayed or irregular visits, lowering their chances of receiving the recommended contents of care to a full extent [43].

Late-initiation ANC visits were identified as significant predictors of missing key elements of care. This is supported by studies conducted in Pakistan [44], Rwanda [39], and Ethiopia [45]. This could be because when women start ANC late, it can be difficult to catch up on missing critical elements of care, such as important tests, screenings, and interventions that are usually done earlier in pregnancy. In addition, commencing ANC in the third trimester may result in less time being available to cover all necessary aspects of care during the remaining visits. Moreover, when women begin ANC late, they may not have the usual healthcare provider or facility for their care. This lack of continuity of care, might result in information gaps and make it more likely that they would miss certain important aspects of care [46].

Similarly, having an inadequate number of ANC visits (only one visit) was significantly associated with missing key care contents. This was in tandem with the studies conducted in Bangladesh [47], Nepal [37], Pakistan [44], and Rwanda [39]. This could be because, with only one ANC visit, healthcare providers have limited time to assess the woman's overall health and perform the relevant screening, tests, education, counselling, and preventive interventions that are usually spread out over numerous visits. It is well recognized that ANC visits allow women to build rapport with their healthcare provider, promoting trust and opening a discussion [9]. This relationship-building aspect is missing in only one visit, which may result in less engagement and compliance with the recommended contents of care.

Women who did not own a mobile phone were more likely to miss the contents of ANC than those who did. Studies conducted in Bangladesh [48], Zanzibar [49], Nigeria [50], and Madagascar [51] support this finding. Women who do not have a mobile phone may have limited access to critical information or updates and may miss appointment reminders or schedule changes. Mobile phones allow people to stay connected and communicate with family, friends, and support networks, which can provide emotional support and aid during the care process [52]. As a result, women without mobile phones may find it more difficult to access their support networks, possibly leading to poor adherence to the contents of care.

Living in a rural part of the country increases the likelihood of missing ANC content. Studies conducted in Bangladesh [47], Vietnam [53], and Ethiopia [45] were in tandem with the current findings. This may be because of a variety of factors. Rural areas in Ethiopia frequently have limited healthcare infrastructure, laboratory setups, and a shortage of healthcare providers. This can lead to long waiting times, delayed appointments, or understaffed clinics, making it challenging for pregnant women to receive comprehensive ANC services. In addition, because poverty is prominent in many rural parts of Ethiopia, pregnant women may face financial difficulties in getting to health facilities on time, and they may prioritize other immediate needs to seek complete ANC contents. Furthermore, owing to a lack of knowledge and information, rural women may be unaware of the potential dangers signs associated with pregnancy; they may have failed to comply with the recommended number of visits and contents of care.

This study had several strengths. First, the results were based on an analysis of nationally representative data, which enhances the generalizability of the findings. Since the findings stem from both spatial and multilevel mixed-effect logistic regression analyses, they can help government and program planners develop geographical, individual, and community-focused public health interventions to minimize the level of missing essential contents of care. However, this finding should be considered in light of some limitations. As the EDHS data are cross-sectional, it is difficult to determine a temporal/causal relationship, and the findings might be exposed to recall- and social desirability biases. In addition, since the study has relied on the secondary data we didn't consider those variables such as waiting time, healthcare worker ethics, and availability of equipment that might increase the odds of missing contents of care.

## Conclusion

The level of missing care content during prenatal visits was high in Ethiopia, with significant spatial variation across regions. Being a rural resident, not having a formal education, commencing ANC late, having inadequate ANC visits, and not having a mobile phone were found to be significant predictors of missing essential contents of care during ANC. Thus, health systems and policymakers should focus on improving access to ANC services, promoting early initiation, and encouraging multiple visits to provide optimal care to pregnant women. In addition, it is essential to focus on enhancing education and healthcare infrastructure in rural countries.

## Supporting information

**S1 Checklist. STROBE statement—Checklist of items that should be included in reports of observational studies.**
(DOCX)

## Acknowledgments

We are grateful to ICF macro (Calverton, USA) for providing the 2016 DHS data of Ethiopia.

## Author Contributions

**Conceptualization:** Aklilu Habte Hailegebireal.

**Formal analysis:** Aklilu Habte Hailegebireal, Habtamu Mellie Bizuayehu.

**Methodology:** Aklilu Habte Hailegebireal, Habtamu Mellie Bizuayehu, Yordanos Sisay Asgedom, Jira Wakoya Feyisa.

**Resources:** Aklilu Habte Hailegebireal.

**Software:** Aklilu Habte Hailegebireal.

**Supervision:** Aklilu Habte Hailegebireal.

**Validation:** Aklilu Habte Hailegebireal.

**Visualization:** Aklilu Habte Hailegebireal.

**Writing – original draft:** Aklilu Habte Hailegebireal, Habtamu Mellie Bizuayehu, Yordanos Sisay Asgedom, Jira Wakoya Feyisa.

**Writing – review & editing:** Aklilu Habte Hailegebireal, Habtamu Mellie Bizuayehu, Yordanos Sisay Asgedom, Jira Wakoya Feyisa.

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
