## [Decision Letter · Decision Letter 0]

5 Feb 2024

PONE-D-23-36880Spatial patterns and predictors of missing key contents of care during prenatal visits in Ethiopia: Spatial and multilevel analysisPLOS ONE

Dear Dr. Habte,

Thank you for submitting your manuscript to PLOS ONE. After careful consideration, we feel that it has merit but does not fully meet PLOS ONE’s publication criteria as it currently stands. Therefore, we invite you to submit a revised version of the manuscript that addresses the points raised during the review process.

We look forward to receiving your revised manuscript.

Kind regards,

Agmasie Damtew Walle

Academic Editor

PLOS ONE

Journal Requirements:

2. We note that Figures 2, 4 and 5 in your submission contain map images which may be copyrighted. All PLOS content is published under the Creative Commons Attribution License (CC BY 4.0), which means that the manuscript, images, and Supporting Information files will be freely available online, and any third party is permitted to access, download, copy, distribute, and use these materials in any way, even commercially, with proper attribution. For these reasons, we cannot publish previously copyrighted maps or satellite images created using proprietary data, such as Google software (Google Maps, Street View, and Earth). For more information, see our copyright guidelines: http://journals.plos.org/plosone/s/licenses-and-copyright.

We require you to either present written permission from the copyright holder to publish these figures specifically under the CC BY 4.0 license, or remove the figures from your submission:

a. You may seek permission from the original copyright holder of Figures 2, 4 and 5 to publish the content specifically under the CC BY 4.0 license.  

Additional Editor Comments:

1. why use Ordinary Kriging interpolation for prediction of missing ANC content

2. what is the reason to select The Bernoulli-based approach was

used to find statistically significant spatial clusters with a high number of women who missed

the content of care

3. please revise the discussion section clearly

4. why not conduct the cluster outlier instead of hotspot analysis

5. please incorporate in introduction section why you want to study this topic in detail

6.include the inclusion and exclusion criteria

7.plese mention what are the gaps that you identified from the previous study

Reviewers' comments:

Reviewer's Responses to Questions

**Comments to the Author**

1. Is the manuscript technically sound, and do the data support the conclusions?

Reviewer #1: Yes

2. Has the statistical analysis been performed appropriately and rigorously? 

Reviewer #1: Yes

3. Have the authors made all data underlying the findings in their manuscript fully available?

Reviewer #1: Yes

4. Is the manuscript presented in an intelligible fashion and written in standard English?

Reviewer #1: Yes

5. Review Comments to the Author

Reviewer #1: - better to include line numbers in the manuscript for smooth communication.

- In the variable subsection, the Authors mentioned "Based on the responses, a composite index was created with minimum and maximum values of 0 and 7, respectively. Finally, the count variable was dichotomized as “1” if any of the seven key contents of care were missed and “0” if she received all the contents of care". Does it scientifically sound to to say 1 if a woman missed only one component? Does it mean missing one component and all the the seven key contents is the same?I see a significant information loss. I think it should be written in a persuasive way to avoid this confusion.

6. PLOS authors have the option to publish the peer review history of their article (what does this mean?). If published, this will include your full peer review and any attached files.

Reviewer #1: No

---

## [Author Response · Author response to Decision Letter 0]

5 Feb 2024

A point-by-point response to editor and reviewers

Authors’ Response to Academic Editor

Dear: Agmasie Damtew Walle, Academic Editor, Plos One

We thank you for a thorough reading and constructive comments and suggestions on our manuscript and for the opportunity to revise and resubmit. We are pleased to submit the revised version of the manuscript titled “Spatial patterns and predictors of missing key contents of care during prenatal visits in Ethiopia: Spatial and multilevel analyses” for your consideration in the special collection of Plos One. The comments of the editors and the reviewers were highly insightful and enabled us to greatly improve the quality of our manuscript. In this revised manuscript we made substantial changes to address your concerns in a point-by-point response. We appreciate your time and look forward to your response and we are very keen to incorporate further comments, if any, for the betterment of the final manuscript.

On the following pages, you will find our responses to the comments and suggestions raised by the esteemed editor and reviewer. 

Sincerely, 

Aklilu Habte(MPH)(corresponding author)

aklilihabte57@gmail.com

Response to Journal requirements

Response: we already prepared the manuscript as per the journal requirement and again we rechecked the compliance towards it during the submission of our revised manuscript.

2. We note that Figures 2, 4 and 5 in your submission contain map images which may be copyrighted. All PLOS content is published under the Creative Commons Attribution License (CC BY 4.0), which means that the manuscript, images, and Supporting Information files will be freely available online, and any third party is permitted to access, download, copy, distribute, and use these materials in any way, even commercially, with proper attribution. For these reasons, we cannot publish previously copyrighted maps or satellite images created using proprietary data, such as Google software (Google Maps, Street View, and Earth). 

You may seek permission from the original copyright holder of Figures 2-9 to publish the content specifically under the CC BY 4.0 license. 

Response: We appreciate your concern to assure the ethical issues. However, all the aforementioned figures(2.4, and 5) in our manuscript are not copyrighted rather they are the result of spatial analysis that we have run in ArcGIS and SaTScan software. The GPS and DHS data that contain Shapefile and other relevant variables were obtained from the DHS office by explaining the objective of the study through online requests. Then, in order to get those figures, we import the relevant data extracted from the 2016 Ethiopian Demographic Health Survey reports and the shapefile of Ethiopia obtained from the 2016 Ethiopian Central Statistical Agency (CSA).To indicate this, we already cited the source of the shapefile alongside each figure. The shape file that we used to construct the figures can be accessed by one of the following links:

1. https://data.humdata.org/dataset/cb58fa1f-687d-4cac-81a7-655ab1efb2d0 .

2. https://gadm.org/download_country.html

Therefore, the maps presented in our study are not copyrighted rather they were the outputs of our spatial analysis results which are the result of those Shapefiles and projected CVS files in ArcGIS. This is the actual procedure that we employed in our present and earlier studies, as well as other Ethiopian researchers. Again we assure you that the figures presented in our study are not copyrighted but rather our spatial analysis results.

Response: Thank you for the suggestion. However, we do not have any supporting information to cite; all of the results were brought up in the manuscript.

Response to Additional Editor Comments: 

Comment 1: why use Ordinary Kriging interpolation for prediction of missing ANC content

Response: Thank you for your insightful inquiry. The possible justifications behind using an ordinary Kriging interpolation were: 

1. It takes into account the spatial dependence or autocorrelation in the data. In the context of ANC content, it recognizes that nearby locations are likely to have similar values. This is important for accurately predicting missing values based on the values observed at surrounding locations.

2. It minimizes the prediction variance by giving more weight to nearby sample points with similar values. This helps in obtaining more reliable predictions, especially in areas where the data is sparse.

3. It provides an optimal linear unbiased estimate of the variable being interpolated by combining the information from neighboring points to estimate the value at a target location while minimizing the bias in the estimation.

4. Furthermore, because ordinary Kriging is based on geostatistical models that account for both spatial trend and spatial variability in data, it provides a more accurate depiction of the underlying spatial patterns in ANC content.

We have added all the possible justifications and highlighted them in the ‘spatial analysis’ section of the "Revised Manuscript with Track Changes" Page 7, Lines 181-187

Comment 2: what is the reason to select The Bernoulli-based approach was used to find statistically significant spatial clusters with a high number of women who missed the content of care.

Response: Thank you for asking this are the reasons for using the model: 

1. The Bernoulli model allows for flexibility in defining the event of interest. In this case, it can be easily adapted to identify spatial clusters of women who missed the content of antenatal care.

2. Furthermore, the results of the Bernoulli-based spatial scan analysis tend to be flexible and easy to interpret, such as revealing clusters with high or lower risk factors, making it easier for policymakers and public health experts to understand and focus interventions.

The details were added and highlighted in the ‘Spatial Scan Statistical Analysis’ section of the "Revised Manuscript with Track Changes" Page 7, Line 195-201

Comment 3: why not conduct the cluster outlier instead of hotspot analysis

Response: Thank you for your insightful inquiry. Hotspot analysis is known to identify areas with statistically significant clusters of high or low values of missing key ANC contents., and it helps in understanding spatial patterns and trends within the data. In addition, Hotspot analysis often involves techniques like Getis-Ord Gi* statistic or Moran's I, which assess the spatial autocorrelation and significance of clustered patterns. On the other hand, cluster outlier analysis focuses on identifying specific locations that deviate significantly from the spatial pattern observed in the dataset.

Our reasons behind prefering Hotspot Analysis are:

• Hotspot analysis is better suited for identifying overall patterns and trends in the data by allowing us to understand where clusters of high or low values are located, which provides a clearer picture.

• When the goal is to inform policy or interventions, hotspot analysis is more appropriate. It helps in identifying areas where targeted efforts or resources may be needed.

• Hotspot analysis provides both global and local insights. Global measures identify overall spatial patterns, while local measures pinpoint specific clusters.

Overall, a hot spot is more preferable one and almost all of the recent studies in Ethiopia and other African countries performed the spatial one.

Comment 4: Please incorporate in the introduction section why you want to study this topic in detail

Response: thank you for your insightful suggestion to incorporate the rationales of the study. Accordingly, we have added the possible reasons behind conducting the current study and highlighted them in the ‘introduction’ section of the "Revised Manuscript with Track Changes" Page 4, Line 102-106

Comment 5: Include the inclusion and exclusion criteria

Response: we appreciate your inquiry. We have incorporated all the eligibility criteria in the ‘Data source and population’ section of the "Revised Manuscript with Track Changes" Page 4, Lines 119-120.

Comment 6: please mention what are the gaps that you identified from the previous study

Response: thank you for your insightful suggestion. The main gap that we identified from previous studies was they primarily focused on the timing and number of visits, and the level of adequacy of services received during visits with their spatial variation has not been thoroughly assessed. Thus, we have mentioned and highlighted this as the main gap and rationale for conducting the current study, in the ‘introduction’ section of the "Revised Manuscript with Track Changes" Page 4, Line 102-106

Comment 7: Please revise the discussion section clearly

Response: Thank you for your suggestion. Accordingly, we thoroughly look into the discussion section again and we have made some amendments. The statements with slight modifications were highlighted throughout the ‘Discussion’ section of the "Revised Manuscript with Track Changes"

Thank you for your constructive comments and suggestions, which we got as valuable input in the improvement of the manuscript. We received all of them as a valuable contribution to our ongoing work. In the following section, we tried to respond to all the possible comments and suggestions from reviewer #1 

END________________________________________

 THANK YOU!!!

Authors’ Response to Reviewer#1 

Dear Reviewer 1, thank you very much for taking the time to review our work and for your positive feedback. We received your thoughtful, and generous review, along with helpful feedback and suggestions, as a valuable contribution to our ongoing work. We have tried to address all the possible comments and suggestions raised by you in the following session.

General comments: 

Comment 1: better to include line numbers in the manuscript for smooth communication.

Response: A sincere apology for not incorporating line numbers for the manuscript and making the review process clattering. As per your suggestion, we have added the line number to the revised version of the manuscript and we gave the responses by mentioning the line and page numbers.

Comment 2: In the variable subsection, the Authors mentioned "Based on the responses, a composite index was created with minimum and maximum values of 0 and 7, respectively. Finally, the count variable was dichotomized as “1” if any of the seven key contents of care were missed and “0” if she received all the contents of care". Does it scientifically sound to to say 1 if a woman missed only one component? Does it mean missing one component and all the seven key contents are the same? I see a significant information loss. I think it should be written in a persuasive way to avoid this confusion.

Response: Thank you for your thoughtful and thorough evaluation. We meant that a woman who missed at least one of the seven items of care was regarded to have missed the contents of care. In contrast, the woman is deemed to have full contents of care only if she received all seven contents of care. The reason behind using such a clear-cut classification is to show the level of the quality of ANC, and that might be the reason the prevalence became high(88.2%).

We have re-write the statement and highlighted it in the ‘Variables of the study’ section of the "Revised Manuscript with Track Changes" Page 5, Line 141-146

Thank you for your constructive comments and suggestions, which we got as valuable input in the improvement of our manuscript. We received all of them as a valuable contribution to our ongoing work. 

END_______________________________________

 THANK YOU!!!

---

## [Decision Letter · Decision Letter 1]

15 May 2024

PONE-D-23-36880R1Spatial patterns and predictors of missing key contents of care during prenatal visits in Ethiopia: Spatial and multilevel analysesPLOS ONE

Dear Dr. Habte,

Thank you for submitting your manuscript to PLOS ONE. After careful consideration, we feel that it has merit but does not fully meet PLOS ONE’s publication criteria as it currently stands. Therefore, we invite you to submit a revised version of the manuscript that addresses the points raised during the review process.

Please note that we have only been able to secure a single reviewer to assess your manuscript. We are issuing a decision on your manuscript at this point to prevent further delays in the evaluation of your manuscript. Please be aware that the editor who handles your revised manuscript might find it necessary to invite additional reviewers to assess this work once the revised manuscript is submitted. However, we will aim to proceed on the basis of this single review if possible. 

We look forward to receiving your revised manuscript.

Kind regards,

Avanti Dey, PhD

Staff Editor

PLOS ONE

Reviewers' comments:

Reviewer's Responses to Questions

**Comments to the Author**

1. If the authors have adequately addressed your comments raised in a previous round of review and you feel that this manuscript is now acceptable for publication, you may indicate that here to bypass the “Comments to the Author” section, enter your conflict of interest statement in the “Confidential to Editor” section, and submit your "Accept" recommendation.

Reviewer #2: (No Response)

2. Is the manuscript technically sound, and do the data support the conclusions?

Reviewer #2: Partly

3. Has the statistical analysis been performed appropriately and rigorously? 

Reviewer #2: Yes

4. Have the authors made all data underlying the findings in their manuscript fully available?

Reviewer #2: Yes

5. Is the manuscript presented in an intelligible fashion and written in standard English?

Reviewer #2: Yes

6. Review Comments to the Author

Reviewer #2: Your final result is influenced by the way you define and measure your variables, so it is important to conduct a reanalysis after making any corrections to your operational definitions.

7. PLOS authors have the option to publish the peer review history of their article (what does this mean?). If published, this will include your full peer review and any attached files.

Reviewer #2: No

---

## [Author Response · Author response to Decision Letter 1]

16 May 2024

A point-by-point response to editor and reviewer

Authors’ Response to Academic Editor

Dear: Avanti Dey, PhD, Staff Editor, Plos One

We thank you for a thorough reading and constructive comments and suggestions on our manuscript and for the opportunity to revise and resubmit. We are pleased to submit the revised version of the manuscript titled “Spatial patterns and predictors of missing key contents of care during prenatal visits in Ethiopia: Spatial and multilevel analyses” for your consideration in the special collection of Plos One. The comments of the editors and the reviewers were highly insightful and enabled us to greatly improve the quality of our manuscript. In this revised manuscript we made substantial changes to address your concerns in a point-by-point response. We appreciate your time and look forward to your response and we are very keen to incorporate further comments, if any, for the betterment of the final manuscript.

On the following pages, you will find our responses to the comments and suggestions raised by the esteemed reviewer. 

Sincerely, 

Aklilu Habte(MPH)(corresponding author)

aklilihabte57@gmail.com

Authors’ Response to Reviewer#1 

Dear Reviewer 1, thank you very much for taking the time to review our work and for your positive feedback. We received your thoughtful, and generous review, along with helpful feedback and suggestions, as a valuable contribution to our ongoing work. We have tried to address all the possible comments and suggestions raised by you in the following session.

Comment 1: Your source data is outdated, from 2016. How important is it for policymakers to consider the changes that have occurred since then? The Ministry of Health has implemented several programs, such as increasing accessibility to laboratories and prioritizing services for

antenatal care (ANC) mothers.

Response: we appreciate your insightful inquiry. We used the 2016 EDHS report since there was no other standard DHS report in Ethiopia, and it was the most recent and relied on a large number of respondents, and the reports from the current study may serve as a basis for evaluating future DHS reports. Initially, we planned to perform the study using the 2019 mini-demographic report, but it lacks the majority of the variables included in the current study, and we were unable to obtain the majority of the contents of care provided during prenatal care. After evaluating both of those challenges, we decided to conduct our analysis on the 2016 report. To obtain the most recent data, we may undertake the study during the upcoming series of standard DHS in 2025/26. Furthermore, the majority of studies that relied on DHS report in Ethiopia used 2016 one. 

Comment 2: The study did not reveal any significant new variables that were previously unknown. What novel information has your study provided?

Response: The following were some of the novel perspectives in the current study.

1. There was a dearth of studies that showed geographical variation in missing adequate contents of care during prenatal visits (inadequacy of care).

2. Previous studies conducted in Ethiopia using DHS reports merely focused on timing(1st Trimester booking) and adequacy of visits(4 or more ANC visits), and they didn’t give due emphasis to missing key contents of care during ANC, which is one of the quality indicators.

3. Also, we tried to incorporate all variables at the individual and community level 

4. Unlike previous studies, this study focused on missing care contents by setting a higher standard; by involving women who missed even one of the necessary seven cares in the category of 'missed care', which may reveal the actual status of ANC quality in the country. 

Comment 3: What suggestions do you have for other researchers?

Response: 

Comment 4: Is it feasible for policymakers to enhance maternal basic education? If so, how can they achieve this?

Response: Yes, they can strengthen basic education in two ways. To begin, they may prioritise female education by increasing adolescent girls school enrollment and reducing dropout rates, which could increase the number of educated women in the population. Second, they can strengthen informal education for women at the community level through community health workers and local administrators, which has already been implemented in several regions of the country. 

Comment 5: Your final result is influenced by the way you define and measure your variables, so it is important to conduct a reanalysis after making any corrections to your operational definitions.

“If a woman missed at least one of the seven items of care, she was considered to have

missed the contents of care(=1), and the woman was only considered to have full contents of care(=0) if she received all seven items of care. 

Response: Thank you very much for letting us make the operational definition more clear for the readers. Accordingly, we have made slight amendments to the operational definition and highlighted it in the ‘Variables of the study’ section of the ‘Revised Manuscript with Track changes’, Lines 141-144, Page 5. As a result, our title, method, results, and discussion were based on the women who missed the contents of care. We politely suggest you to look at the manuscript.

Comment 6: Have you taken into account other factors such as waiting time, healthcare worker ethics, and so on?

Response: thank you for asking. However, the study relied on the EDHS data and we didn’t have any opportunity to consider those variables. Thus, we added it as the limitation of this study in the ‘Discussion’ section of the ‘Revised Manuscript with Track changes’, Lines 432-435, Page 18.

Comment 7: What is the clear gap identified in the previous study?

Response: These were some of the gaps: 

1. They didn't show geographical variation in missing adequate contents of care during prenatal visits (inadequacy of care),

2. They merely focused on timing(1st Trimester booking) and adequacy of visits(4 or more ANC visits), and they didn’t give due emphasis to missing key contents of care during ANC, which is one of the quality indicators, and 

3. They also failed to show which content of care was missed by most women and in which region in the way that we presented in Table 4. Thus we tried to address those gaps in this study. 

Comment 8: Further elaboration is required in the discussion section, as well as a more comprehensive exploration of the limitations.

Response: thank you for your suggestion. Accordingly, we tried to add some elaborations and possible justifications in the discussion section and we highlighted them in the ‘Revised Manuscript with Track changes’. In addition, we add another limitation in the ‘Discussion’ section of the ‘Revised Manuscript with Track changes’, Lines 432-435, Page 18.

Thank you for your constructive comments and suggestions, which we got as valuable input in the improvement of our manuscript. We received all of them as a valuable contribution to our ongoing work. 

END_______________________________________

 THANK YOU!!!

---

## [Decision Letter · Decision Letter 2]

4 Nov 2024

Spatial patterns and predictors of missing key contents of care during prenatal visits in Ethiopia: Spatial and multilevel analyses

PONE-D-23-36880R2

Dear Dr. Hailegebireal,

We’re pleased to inform you that your manuscript has been judged scientifically suitable for publication and will be formally accepted for publication once it meets all outstanding technical requirements.

Kind regards,

Laura Kelly, PhD

Division Editor

PLOS ONE

Additional Editor Comments (optional):

Reviewers' comments:

Reviewer's Responses to Questions

**Comments to the Author**

1. If the authors have adequately addressed your comments raised in a previous round of review and you feel that this manuscript is now acceptable for publication, you may indicate that here to bypass the “Comments to the Author” section, enter your conflict of interest statement in the “Confidential to Editor” section, and submit your "Accept" recommendation.

Reviewer #2: All comments have been addressed

2. Is the manuscript technically sound, and do the data support the conclusions?

Reviewer #2: Yes

3. Has the statistical analysis been performed appropriately and rigorously? 

Reviewer #2: Yes

4. Have the authors made all data underlying the findings in their manuscript fully available?

Reviewer #2: Yes

5. Is the manuscript presented in an intelligible fashion and written in standard English?

Reviewer #2: Yes

6. Review Comments to the Author

Reviewer #2: I would like to express my gratitude for taking the time to address my comments thoroughly. Having reviewed the responses and considering the adjustments made, I find that I have no further comments to add at this time

7. PLOS authors have the option to publish the peer review history of their article (what does this mean?). If published, this will include your full peer review and any attached files.

Reviewer #2: No

---

## [Editor Report · Acceptance letter]

7 Nov 2024

PONE-D-23-36880R2 

PLOS ONE

Dear Dr. Hailegebireal, 

I'm pleased to inform you that your manuscript has been deemed suitable for publication in PLOS ONE. Congratulations! Your manuscript is now being handed over to our production team.

Kind regards, 

on behalf of

Dr. Laura Hannah Kelly 

Staff Editor

PLOS ONE